# The dynamic genetic determinants of increased transcriptional divergence in spermatids

Jasper Panten[1,2,3], Tobias Heinen[2,4,5], Christina Ernst [6], Nils Eling [7,8], Rebecca E. Wagner[3,9], Maja Satorius[1], John C. Marioni [10,11,12], Oliver Stegle [2,5] ✉ & Duncan T. Odom [1,3,10] ✉

*Cis*-genetic effects are key determinants of transcriptional divergence in discrete tissues and cell types. However, how *cis*- and *trans*-effects act across continuous trajectories of cellular differentiation in vivo is poorly understood. Here, we quantify allele-specific expression during spermatogenic differentiation at single-cell resolution in an F1 hybrid mouse system, allowing for the comprehensive characterisation of *cis*- and *trans*-genetic effects, including their dynamics across cellular differentiation. Collectively, almost half of the genes subject to genetic regulation show evidence for dynamic *cis*-effects that vary during differentiation. Our system also allows us to robustly identify dynamic *trans*-effects, which are less pervasive than *cis*-effects. In aggregate, genetic effects were strongest in round spermatids, which parallels their increased transcriptional divergence we identified between species. Our approach provides a comprehensive quantification of the variability of genetic effects in vivo, and demonstrates a widely applicable strategy to dissect the impact of regulatory variants on gene regulation in dynamic systems.

The comprehensive characterisation of the impact of DNA sequence changes on molecular traits that ultimately drive phenotypic variation remains an unresolved challenge. Expression quantitative trait locus (eQTL) mapping in bulk tissues has revealed extensive tissue-[1,2] and cell type-specificity[3] of regulatory variants. Most recently, advances in single-cell sequencing have enabled genetic regulation to be probed at cellular resolution, revealing extensive context-dependence of regulatory variants also between more subtle cellular subtypes[4–10]. Furthermore, genetic analyses in in vitro differentiation systems using human pluripotent stem cells have revealed dynamically changing *cis*-effects across continuous differentiation processes[4,5,7,10]. These studies highlight the complexity at which regulatory dependencies are affected by cell type transitions, which has important implications for human physiology and disease.

However, our ability to comprehensively identify and characterise the dynamics of regulatory variants remains limited due to shortcomings of existing models. In particular, analysis of *trans*-eQTL using classical population approaches has limited power due to the

[1]Division of Regulatory Genomics and Cancer Evolution, German Cancer Research Centre (DKFZ), 69120 Heidelberg, Germany. [2]Division of Computational Genomics and Systems Genetics, German Cancer Research Centre (DKFZ), 69120 Heidelberg, Germany. [3]Faculty of Biosciences, Heidelberg University, 69117 Heidelberg, Germany. [4]Faculty of Mathematics and Computer Science, Heidelberg University, Heidelberg, Germany. [5]European Molecular Biology Laboratory, Genome Biology Unit, 69117 Heidelberg, Germany. [6]School of Life Sciences, Ecole Polytechnique Fédérale de Lausanne (EPFL), 1015 Lausanne, Switzerland. [7]University of Zurich, Department of Quantitative Biomedicine, Zurich 8057, Switzerland. [8]ETH Zurich, Institute for Molecular Health Sciences, Zurich 8093, Switzerland. [9]Division of Mechanisms Regulating Gene Expression, German Cancer Research Centre (DKFZ), 69120 Heidelberg, Germany. [10]Cancer Research UK Cambridge Institute, University of Cambridge, Cambridge, UK. [11]European Molecular Biology Laboratory, European Bioinformatics Institute, Wellcome Genome Campus, Cambridge, UK. [12]Wellcome Sanger Institute, Wellcome Genome Campus, Cambridge, UK. ✉e-mail: o.stegle@dkfz-heidelberg.de; d.odom@dkfz-heidelberg.de

prohibitively large sample sizes required[11] and even for *cis*-effects, it can be challenging to identify the dynamics of context-dependent genetic effects in an unbiased manner. Furthermore, in vitro models cannot perfectly recapitulate the complexity of cellular development in vivo. We therefore sought to develop an experimental system that allows for the comprehensive and quantitative analysis of the context-dependency of genetic effects in vivo.

Here, we leverage a classical F1 hybrid system combined with single-cell RNA-Sequencing[12–16] to assay both dynamic *cis*- and *trans*-genetic effects in an unbiased manner. We analyse male germ cell development as a model of cellular differentiation, which features continuous, unidirectional cell type transitions and, compared with other tissues, a faster accumulation of species-specific gene expression changes[17–19]. Indeed, much of this higher transcriptional divergence appears concentrated in spermatids, and may result from reduced constraint on gene expression levels or increased positive selection[19–21]. Our single-cell based approach expands on previous efforts to quantify *cis*- and *trans*-effects on gene expression in testes using bulk RNA-seq profiling, which were limited to discrete cell types[22,23].

Our data reveal that at least 40% of *cis*-effects exhibit a dynamic component, varying significantly across differentiation. Dynamic *trans*-effects exist in spermatogenesis, but exert only a minor impact on allele-specific expression. We finally show that among male germ cell types, dynamic genetic effects are most common in round spermatids, paralleling their increased transcriptional divergence.

## Results

### Identifying the cell type-specificity of *cis*- and *trans*-acting genetic effects across mouse spermatogenesis

We propose an approach to assay dynamic regulatory effects acting in *cis* and *trans* by combining a classical F1 genetic design with single-cell RNA-sequencing across the continuous trajectory of sperm development[2,14,24,25] (Fig. 1). First, the F1 cross allows the quantification of the *cis*- and *trans*-driven components of strain-specific gene expression by placing both alleles in the same nuclear environment. In the F1 mouse, any bias measured between the alleles will therefore only be due to genetic *cis*-effects. In contrast, expression differences between the F0 strains (for simplicity, we will also refer to this as allele-specific expression) are impacted by both classes of regulatory effects (*cis* and *trans*). Therefore, *trans*-effects can be measured as differences in allelic balance between F0 and F1 mice. Secondly, RNA-sequencing based profiling provides information about cell type identity of single cells and their pseudotemporal ordering, which allows for identifying dynamic changes of *cis* and *trans* effects across the differentiation trajectory.

We applied this experimental strategy to profile single-cell transcriptomes of testes from male *Mus musculus* (C57BL/6-Ly5.1), *Mus Castaneus* (CAST/EiJ), and their F1 offspring in six biological replicates using 10X Genomics 3′ scRNA-seq protocols (18 experiments in total, 1202−4361 cells per experiment) (Methods). Using an integrated dataset of 54,863 cells (after quality control, Methods, Fig. S1a–d), we captured the complete trajectory of sperm development, identifying 4 major germ cell and 3 somatic cell types, as expected[18,26–28] (Figs. 2b, S1e). The high density of genetic variation between C57BL/6 and CAST/EiJ enabled allelic quantifications of RNA abundance in individual cells from F1 mice, thus providing allelic resolution for 25.82% of reads and 54.98% (6495 / 11,812) of all expressed genes (>50 allele-specific reads per sample, Methods). Validating our allelic quantifications, our mapping strategy showed the expected strong maternal allelic bias in mitochondrial and X-linked genes in the F1 data (Fig. S1h-k, o). Quantification of allelic resolution among the haploid post-meiotic spermatids was possible because these cells share RNA through cytoplasmic bridges and therefore contain RNA from both alleles, similar to diploid cells (Fig. S1l[29]). Overall, sperm development in all three genetic

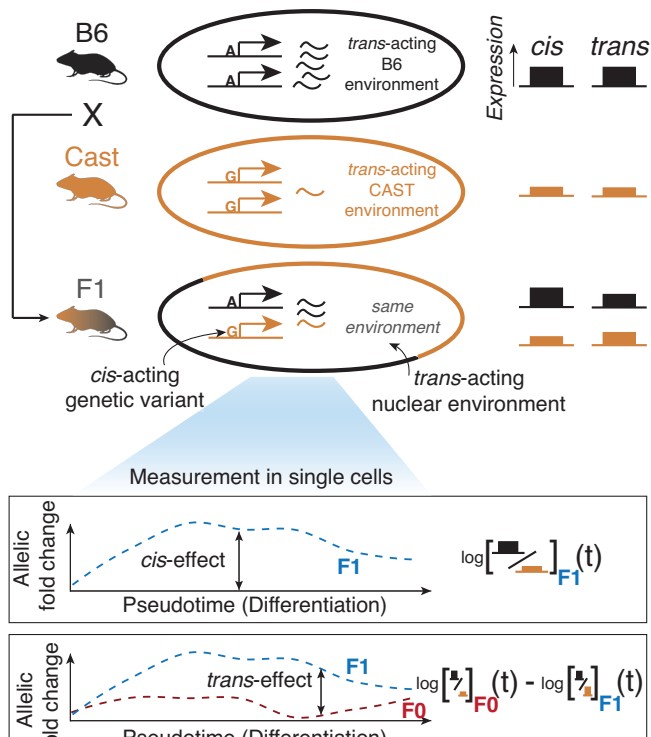

**Fig. 1 | Overview of the experimental strategy.** Comparison of allele-specific expression in single cells between parental strains and their F1 crosses can reveal context-dependent *cis*- and *trans*-effects. The mouse and testis icons were created with Biorender.com.

backgrounds showed highly similar cell type proportions and transcription patterns (Fig. S1d-g), allowing for direct comparison of allelic ratios to analyse context-specific *cis*- and *trans*-contributions.

We first sought to verify that we can identify genetic effects in pseudo-bulk aggregate samples of either the whole tissue or individual germ cell types, using single-cell RNA-sequencing of the F0 parents and F1 crosses, as previously shown using bulk RNA sequencing[14]. We adapted existing modelling strategies of F0 and F1 data based on negative- and beta-binomial distributions[14] to estimate *cis*- and *trans*-effects acting on individual genes (Methods). First, using whole tissue aggregates, we assigned the regulatory classes (that is, whether it is *cis*- or *trans*-acting) of 3230 genes (out of 6495 genes) to be either *cis*-acting, *trans*-acting, or a combination of both (Fig. 2c). This magnitude is similar to other homeostatic mammalian tissues[14,30] (Fig. S2). Second, we assigned regulatory classes separately for major cell types, including spermatocytes (2031 genes), round spermatids (3025 genes), and elongating spermatids (2158 genes) (Fig. 2d). This cell-type specific analysis allowed us to assign a regulatory class to a total of 3349 genes, improving on the 3189 genes obtained from the analysis of all cells at once (Fig. S1m,n).

To assess the magnitude of the expression changes, we also quantified the fold change effect for genes in these categories, finding that 22% − 35% had absolute log allelic fold changes > 1 with *cis* effects generally being larger than *trans*-effects (13%-28% across populations) (Fig. S3a,b). Furthermore, our ability to detect genetic effects is not primarily driven by expression level (Fig. S3d) and we observed similar sets of genes when comparing our model-based approach to conventional differential expression analysis between the founder strains (Fig. S3c). Next, we tested for significant differences in *cis*-acting regulation between cell types, by assessing differential allelic effects in the F1 populations for individual genes (generalised linear model; Methods).

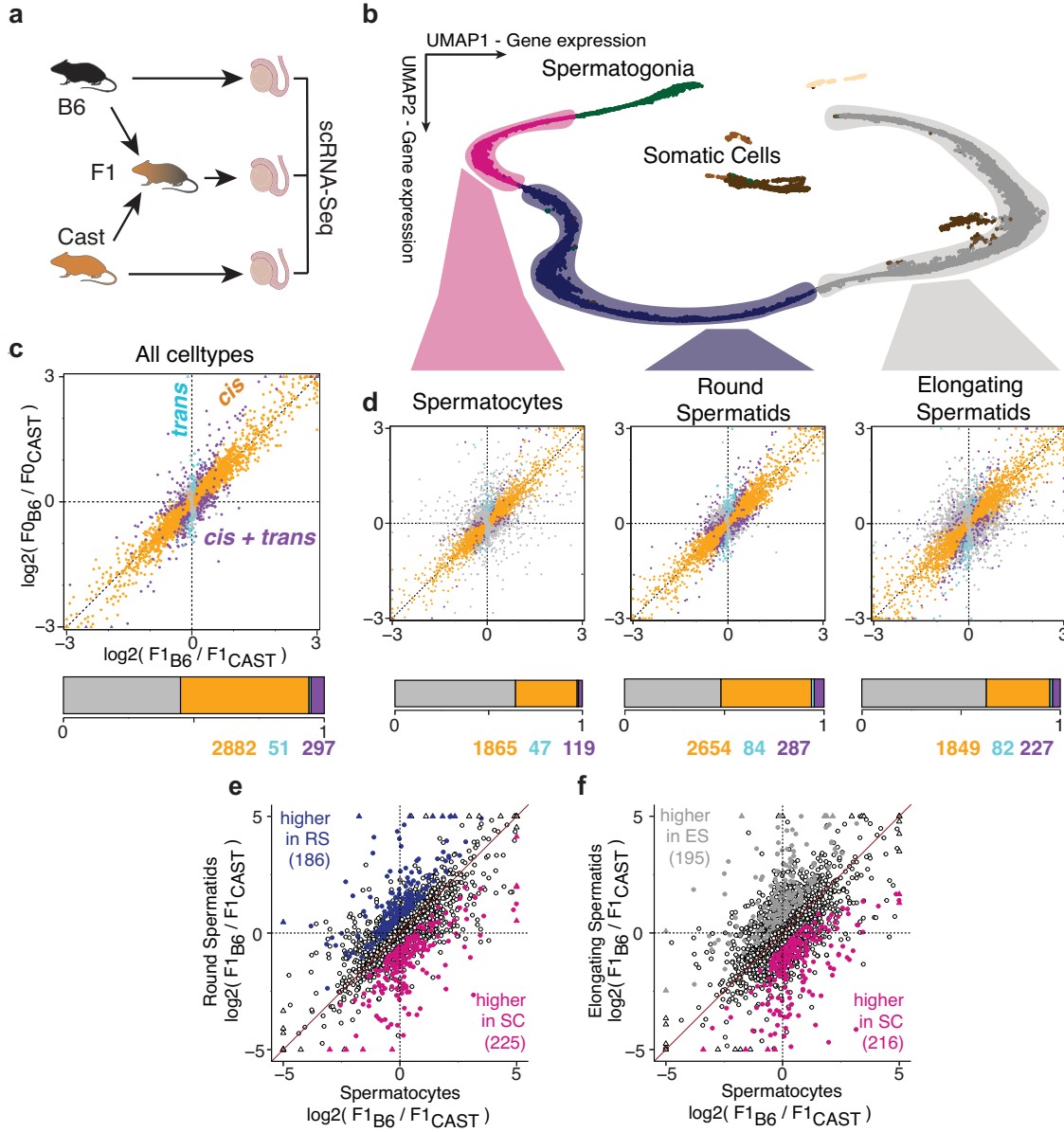

**Fig. 2 | Quantification of cell type specificity of *cis*- and *trans*-acting genetic effects on gene expression across mouse spermatogenesis using single-cell RNA sequencing. a** Testes from eight-twelve week old mice were subjected to 10x Genomics scRNA-Seq profiling in 6 replicates for the F1 offspring from a C57BL/6-Ly5.1 (female) x CAST/EiJ (male) cross, and for their parental strains. The mouse and testis icons were created with Biorender.com. **b** UMAP representation of 7895 F1 cells from two replicates with colour denoting major cell type annotation. **c** Top: Scatter plot comparing log fold changes in gene expression from B6 and CAST alleles in F1 mice (x-axis) and gene expression between B6 and CAST alleles in parental mice (y-axis), quantified in aggregate across all cells. Colours denote the classification of regulatory mechanisms for each gene (*conserved, cis, trans, cis + trans*; Methods). Triangles indicate genes with absolute values larger than 3. Bottom: Barplot denoting the proportion of genes in each category. **d** As (**c**), however, consider aggregate expression estimates in the three major cell types. **e**, **f** Scatter plots between log2 allelic fold changes log2 (B6 / CAST) in spermatocytes versus (**e**) round spermatids and (**f**) elongating spermatids. Genes marked in colour have differential allelic imbalances between cell types (adjusted p-value < 0.1; generalised linear model; Methods). Triangles indicate genes with absolute values larger than 5. Analyses in (**c**)–(**f**) are based on genes with at least 50 allelic reads per sample (6495 genes).

This identified 411 genes with differential *cis* effects between spermatocytes and round spermatids (adjusted p-value < 0.1, absolute difference in log2 aFC between the two cell types > 0.5), and 411 genes with differential effects between spermatocytes and elongating spermatids (adjusted p-value < 0.1, absolute difference in log2 aFC > 0.5) (Fig. 2e, f). Furthermore, dimensionality reduction of cells based on their allelic imbalances recreated the continuous trajectory of sperm differentiation (Fig. S1p, Methods), initially generated using gene expression (Fig. 2b).

In summary, single-cell mapping of sperm development in F1 mice readily assigns *cis*- and *trans*-effects in major cell types of spermatogenesis, revealing substantial cell-type specificity in genetic regulation.

## Dynamic changes in *cis*-acting genetic effects on transcription across sperm differentiation
Next, we used an analysis approach that does not rely on the definition of discrete cell types, but instead leverages the continuous nature of sperm development assayed using scRNA-seq to model the underlying regulatory mechanisms. To this end, we first defined spermatogenic differentiation across F0 and F1 germ cells as a joint pseudotemporal ordering (Methods, Fig. S4a,b). We then applied a recently developed

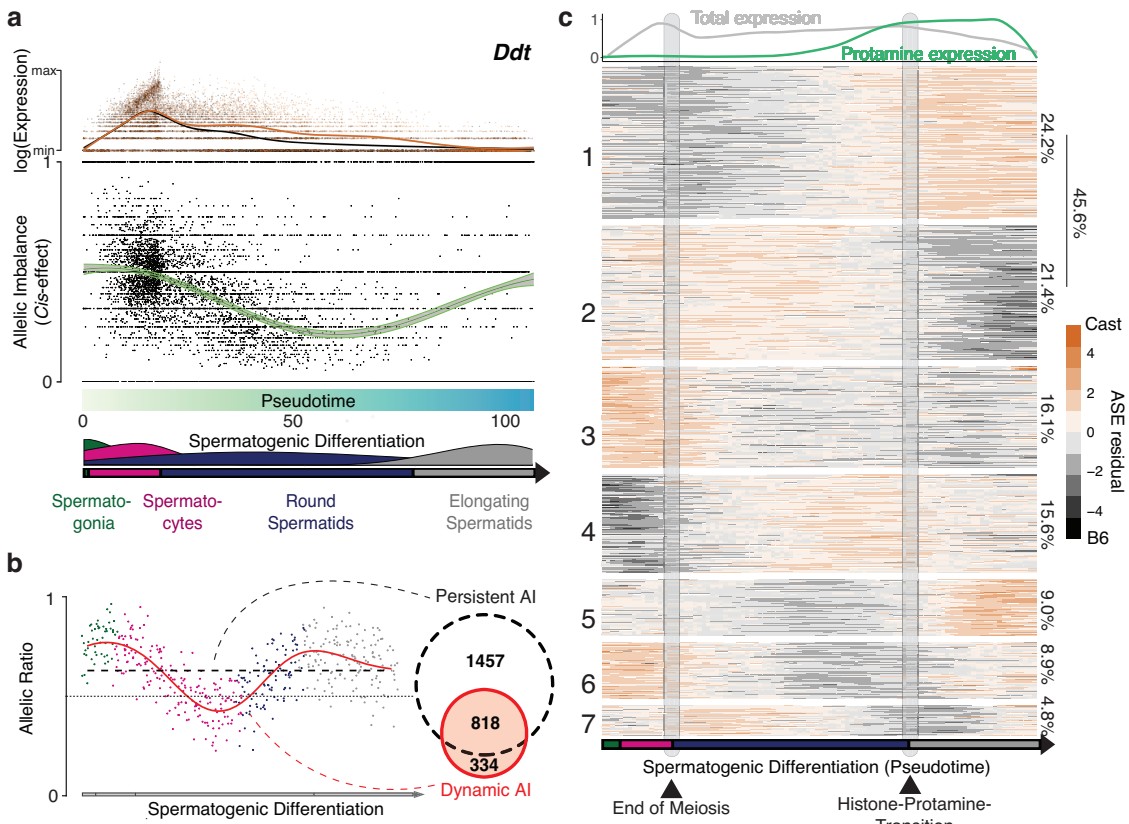

**Fig. 3 | Widespread dynamic changes in *cis*-acting genetic effects on transcription across sperm differentiation in F1 mice. a** Top: Relative gene expression of *Ddt*, across sperm differentiation (x-axis), either quantified for the B6 (black) or the CAST (orange) allele. Dots correspond to allelic expression quantifications in individual cells; solid lines correspond to interpolated trajectories (LOESS fit). Middle: Allelic imbalance ratios B6 / (B6 + CAST) across individual cells as in the top panel. Solid lines correspond to the estimated latent trajectory (scDALI fit) with shaded areas denoting plus or minus two standard deviations (of the latent trajectory). Bottom: Pseudotime ordering used in the top and middle panel with associated cell type assignments across spermatogenic differentiation. **b** Left: Illustration of the scDALI model for allelic imbalance, which decomposes allelic effects into a persistent (block horizontal line; agnostic to differentiation stage) and

a dynamic (red line; scDALI interpolation, variable across differentiation) component of allelic imbalance. Both types of allelic imbalance can be analysed. Right: Venn diagram showing the number of genes with evidence for persistent and/or dynamic allelic imbalance (adjusted p-value < 0.01; scDALI test; Methods). **c** Heatmap of z-transformed scDALI-interpolated allelic trajectories across sperm differentiation for 709 genes with evidence for dynamic allelic imbalance. The x-axis represents 100 evenly spaced sampling points. Genes are grouped into 7 clusters using hierarchical clustering of their allelic imbalance trajectories. Vertical bars indicate the meiosis and histone-to-protamine transitions as derived from cell type annotations. Top panel shows smoothed total and protamine expression (average expression of *Tnp1*, *Tnp2*, *Prm1* and *Prm2*).

analysis framework based on Gaussian process regression (scDALI[31], Fig. 3a) to identify patterns of allelic imbalance across the differentiation time course. We specifically used scDALI to assess the evidence for two alternative *cis*-acting modes shaping allelic balance: (1) persistent, where the ratio of expression between the two alleles is unchanged across differentiation, and/or (2) dynamic, where the allelic ratio changes as a function of the cell state (Fig. 3b, Methods)[5,31].

scDALI applied to 4039 genes in which allelic expression could be robustly quantified (>1000 allelic reads; adjusted p-value < 0.01, Fig. 3b left panel), identified 2275 genes with persistent allelic imbalance in their transcription (adjusted p-value < 0.01, Fig. 3b). Additionally, the scDALI test for dynamic *cis*-effects identified 1152 genes with evidence for variation in allelic ratios as a function of the cell state (adjusted p-value < 0.01). Notably, 818 genes had evidence for both a dynamic and a static regulatory model, and 334 were exclusively identified as dynamically acting (Figs. 3b, S4c). We also compared our continuous modelling approach to applying scDALI to discrete cell type classes, dividing spermatogenesis into 4 cell types, which revealed markedly fewer genes with evidence for cell-type specific allelic imbalance (Fig. S4d–f).

Next, we used hierarchical clustering to identify groups of genes with shared patterns of allelic dynamics across sperm development

(Methods, Fig. 3c). This identified seven clusters, which are characterised by punctuated changes in the transcriptional balance between alleles, often coinciding with developmental transition points. Changes in dynamic allelic imbalance are not driven by changes in total transcription levels[5] (Fig. S4g–h). The two largest clusters (clusters 1 and 2; collectively covering 45.61% of all genes) showed mirror-image changes in allelic imbalance, specifically at the histone-protamine transition. The histone-to-protamine transition silences the spermatid genome, and is one of the most extreme alterations in DNA structure and compaction found in eukaryotes[32,33]. The remaining five clusters showed additional major changes in allelic balances, for example at the transition out of meiosis, a similarly extensive remodelling of the genomic regulatory landscape.

We considered whether dynamic *cis*-effects could result mainly from RNA degradation. If so, then we reasoned that allelic imbalance should be strongest during gene down-regulation. To this end, we identified 726 genes that are both up- and down-regulated during differentiation. We then asked for each of these genes when the allelic imbalance was strongest: in the beginning, middle or end of the differentiation trajectory. We found that genes peaked in allelic imbalance across all stages of up- and down-regulation, and 20% showed strongest allelic bias during gene up-regulation. Together, this

indicates that allele-specific RNA stability is not the sole factor driving allelic imbalance (Fig. S5a–d).

We also tested whether in spermatids, allelic imbalance in gene expression might be caused by their haploid genomic state. Contrary to the expectation of a haploid cell, we observed biallelic expression at chromosome scale, similar to spermatocytes and spermatogonial stem cells, consistent with cytoplasmic sharing of most transcripts (Fig. S1l). The subset of genes reported to resist cytoplasmic sharing during spermiogenesis[29] were not enriched with persistent or dynamic *cis*-effects, indicating that genetic effects, not incomplete sharing of RNA, drive allelic imbalance (Fig. S6).

We next asked whether allele-specific expression could be explained by corresponding differences in chromatin accessibility. To this end, we isolated tetraploid spermatocytes using fluorescence-activated cell sorting, performed (ATAC-Seq) and measured allele-specific chromatin accessibility (asCA) at 6844 sites across 3096 genes for which we measured allele-specific expression (Fig. S7a–c). We observed that genes with allelic effects in expression were enriched for genes with asCA across a range of allelic effect sizes (Fig. S7d, e). Furthermore, we observed that allele-specific expression close to genes with asCA (in spermatocytes) was strongest in spermatocytes, suggesting that dynamic *cis*-effects can be driven by cell type-specific changes in chromatin accessibility (Fig. S7f, g).

In summary, our analyses reveal that almost half of genes with allelic bias exhibit dynamic changes in allelic balance across sperm differentiation. The remodelling of the genome during meiosis and protamine deployment is associated with large-scale and *cis*-directed changes in the allelic contributions to gene expression. Finally, our results suggest that allelic imbalance during spermatogenesis can result from both allele-specificity in transcriptional regulation or transcript stability.

## Identification of context-dependent *trans*-effects across cellular differentiation in vivo

Similarly, our F1 hybrid system can identify persistent and dynamic *trans*-effects, which manifest as differences in allelic imbalance between parental and F1 mice[13,14]. In contrast, comprehensive mapping of dynamic *trans*-effects in population-based eQTL studies would demand sample sizes that are currently inaccessible to single-cell approaches[11]. We first incorporated the single-cell data from the parental strains (F0) into our analysis by first establishing a common coordinate system across all strains, again leveraging the pseudo-temporal ordering (Methods, Fig. S4a,b). We sorted the cells into 100 temporally ordered bins in which we estimated total expression ratios in the F0 parental strains as well the corresponding allelic imbalance in their F1 offspring. The change in the differences between these ratios allows for quantification of dynamic changes in the *trans*-component of gene regulation (Fig. 4a).

We then extended the scDALI framework to jointly model the allelic expression trajectories in the parental strains and their F1 offspring to systematically discover genes with dynamic *trans*-effects. This integrated approach enables the evidence for both persistent (constant non-zero difference between F0 and F1) and dynamic *trans*-components (non-trivial covariance between F0 and F1) to be assessed using Bayes Factors (Fig. 4b, Methods). We calibrated the Bayes Factor threshold to compare *cis*- and *trans*-effects by setting it to match the number of genes with dynamic *cis*-effects identified using scDALI (adjusted p-value < 0.0001 corresponding to log BF > 10; Fig. S10a,b, Methods). Applied to 3657 genes, this model identified genes with evidence for persistent and dynamic *trans*-effects, contributing to the transcriptional differences between CAST and B6 alleles of 352 (9.6%) and 117 (3.2%) of genes, respectively (log BF > 10) (Fig. 4c). To our knowledge, the *trans*-acting components during mammalian differentiation have never been comprehensively analysed within individual cell types in vivo (see also[2]).

To provide additional confidence that persistent and dynamic *trans*-effects identified by our model are genuine, we carried out a number of additional analyses, including evaluation of *trans*-effect size estimates, the positional distribution of *trans*-effects across differentiation, and the relationship between expression level and *trans*-regulation (Fig. S10d–f).

Similar to the genes exhibiting *cis*-effects (Fig. S8), genes with evidence for *trans*-components appeared to be under reduced selective pressure (Fig. 4d). Notably, a similar fraction of *trans*-effects was dynamic (27.2%) compared to *cis*-effects (30.7%, Fig. 4e), which indicates that *trans*-effects are not more likely to be influenced by the specific cellular differentiation state than *cis*-effects. Finally, *cis*- and *trans*-effects appear to be largely independent, based on our assessment of the evidence for their co-occurrence for either persistent (Fig. 4f) or dynamic (Fig. 4g) regulation of the same genes.

In summary, we provide a framework to identify cell type-specific *trans*-effects from F1 and parental data and show that both static and dynamic *trans*-effects exist, but are rare.

## Within-species allelic imbalance corresponds to between-species transcriptional divergence

In whole steady-state tissues, *cis*-effects are the main driver of transcriptional divergence[14,30,34]. However on a single-cell basis, how the dynamics of how *cis*- and *trans*-linked variation contributes to gene expression divergence during differentiation is not known.

We first identified genes with evidence for dynamic transcriptional divergence between the parental strains (total: 924 genes, Fig. 5a, Methods). Strikingly, these genes were enriched for dynamic *cis*-effects (63.6% *cis* vs 12.0% of *trans*, Fig. 5d). Next, we asked whether regulatory variation in *cis* and gene expression changes follow the same trajectories over differentiation. Across sperm differentiation, we employed a joint hierarchical clustering approach to identify groups of genes with a common pattern of dynamic *cis*- and *trans*-effects, as well as transcriptional divergence (Fig. 5a–c). Most genes were in clusters that peak in round spermatids. For instance, clusters 1, 3, 5 and 6 peak in round spermatids, and together correspond to 61.9% of dynamic *cis*-effects and 64.9% of dynamic divergent genes. Moreover, at the level of individual genes, transcriptional divergence and cis-effects on allelic imbalance follow highly similar trajectories, suggesting that expression divergence is largely driven by *cis*-effects (Fig. 5e).

We eliminated the possibility that the variation in total expression levels of individual genes could explain the dynamics of allelic variation (Fig. S11a,b). Although dynamic *trans*-effects were comparatively few in number (117 high confidence effects, Fig. 4e), they similarly accumulated late in differentiation (Fig. 5c). The predominance of allelic effects and expression divergence in round spermatids was also apparent in transcriptome-wide aggregated effects (Fig. 5b, c, bottom AES insets).

In sum, during murine spermatogenesis, dynamic genetic effects are mainly driven by *cis*-acting variants, which contribute to the increased transcriptional divergence in round spermatids compared to other cell types.

## The concentration of expression divergence in round spermatids is shared between species

We identified dynamic *cis*-acting regulatory variants as a driver of transcriptional divergence between *Mus domesticus* and *Mus castaneus*, with a striking concentration in round spermatids (Fig. 5a–e). We asked to what extent this mechanism generalises to other mouse species, as has been suggested for primates and other vertebrates[17,19,20].

We therefore generated an additional and independent dataset of matched single-cell RNA-seq maps using testes from 2 novel replicates of B6 and CAST, as well as from *Mus caroli* as a third, highly divergent

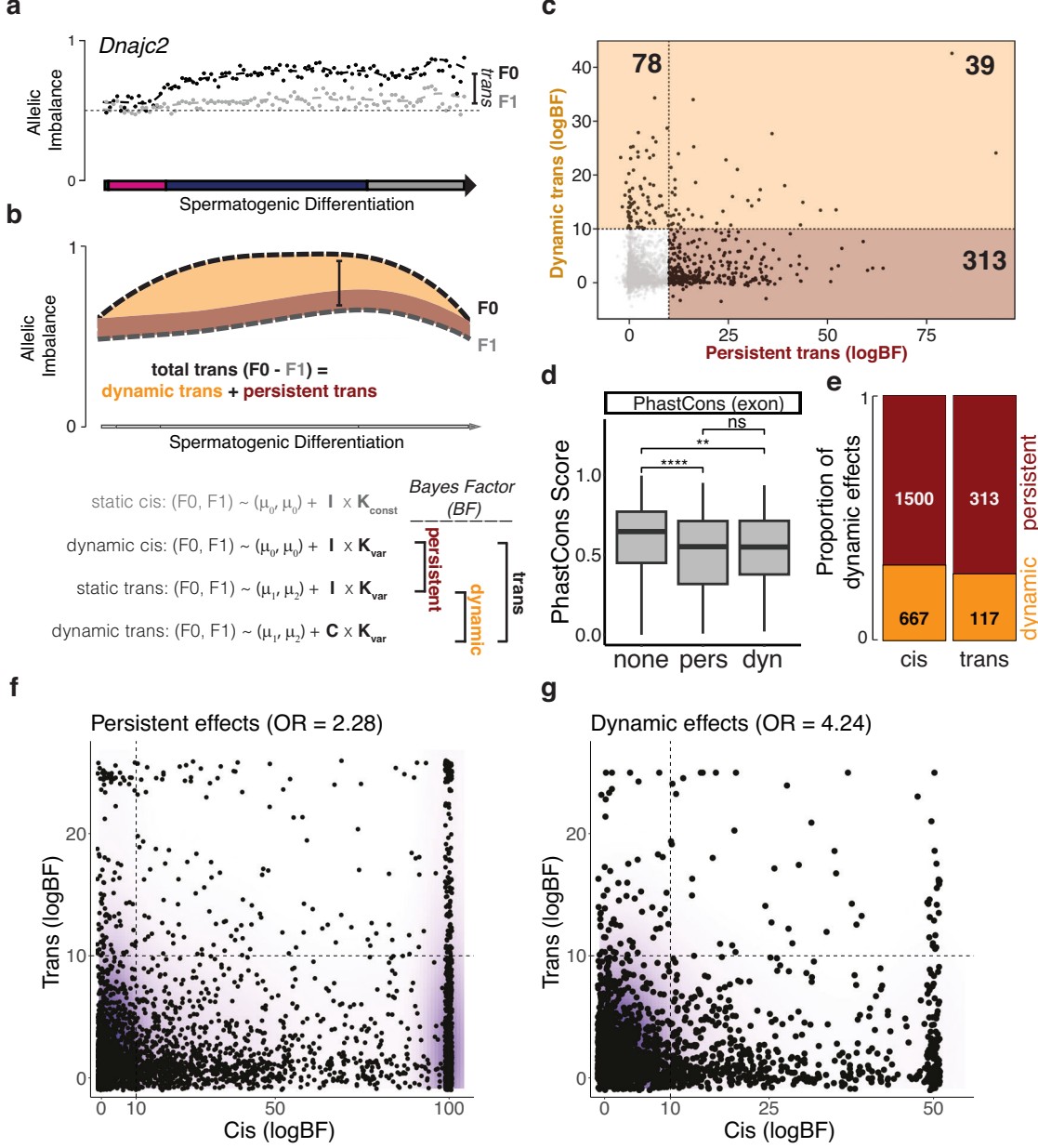

**Fig. 4 | Differential allelic dynamics between parents and F1 offspring reveal context-dependent *trans*-effects in spermatogenesis. a** Interpolated trajectories of allelic imbalance in F0 parental mice (black) and F1 (grey) offspring for *Dnajc2*. Dots denote average allelic ratios from cells distributed in 100 evenly spaced bins across the differentiation trajectory. **b** Graphical (top) and mathematical (bottom) representation of the analysis strategy to identify different types of *trans*-effects. Latent allelic trajectories are modelled by two non-linear functions derived from Gaussian process regression (Methods). Persistent *trans*-effects manifest as a persistent difference between F0 and F1 allelic ratios (red), dynamic *trans*-effects as a dynamic difference (orange). Evidence for either type of effect is evaluated using Bayes Factors (BF). **c** Scatterplot of Bayes Factors that correspond to persistent versus dynamic *trans*-effects. Horizontal and vertical lines correspond to a logBFs > 10 threshold; the number of genes with evidence for persistent and/or dynamic effects are highlighted. **d** Box plots of exonic PhastCons scores for genes with no (*n* = 3227 genes), persistent (*n* = 313) and dynamic (*n* = 117) *trans*-effects (Two-sided Wilcoxon's rank sum test, ****=6.6e-09, **=0.0014). The boxplots show median, 25%- and 75%-quantiles, the whiskers 1.5 inter-quartile ranges. **e** Comparison of the fraction of dynamic versus persistent *cis*- and *trans*-effects (log Bayes Factor cutoff = 10, set to match the number of dynamic *cis* effects identified using scDALI; Methods). *p* = 0.025 using a two-sided Chi-square test. **f** Scatterplot between log Bayes Factors for persistent and *cis*- versus persistent *trans*-effects. Odds ratio of co-occurrence of both effects for the same genes based on logBF threshold 10. **g** Scatter plot as in (**f**), however for dynamic *cis*- versus dynamic *trans*-effects.

---

mouse species (CAROLI/EiJ) in order to quantify the transcriptional divergence across spermatogenesis (Fig. 6a,b; Fig. S12a–g). The evolutionary proximity of these three species allowed us to accurately project individual cells onto the previously-defined pseudotime coordinates (Fig. S12h,i; Methods). The resulting integrated dataset featured transitions in cell type proportions and marker gene expression across differentiation consistent with our prior F1 dataset (Fig. S12f,g). We validated that differential gene expression between B6

and CAST was highly correlated between these two independently collected datasets (Fig. S13).

Next, we compared the data from CAST and CAROLI, versus C57BL/6 as a reference, and clustered divergent genes by their temporal similarity across spermatogenesis. Although on the level of individual genes, there were substantial differences between species (Fig. S12j,k), the overall dynamics of expression divergence, again, concentrated late in sperm differentiation (Fig. 6d, e).

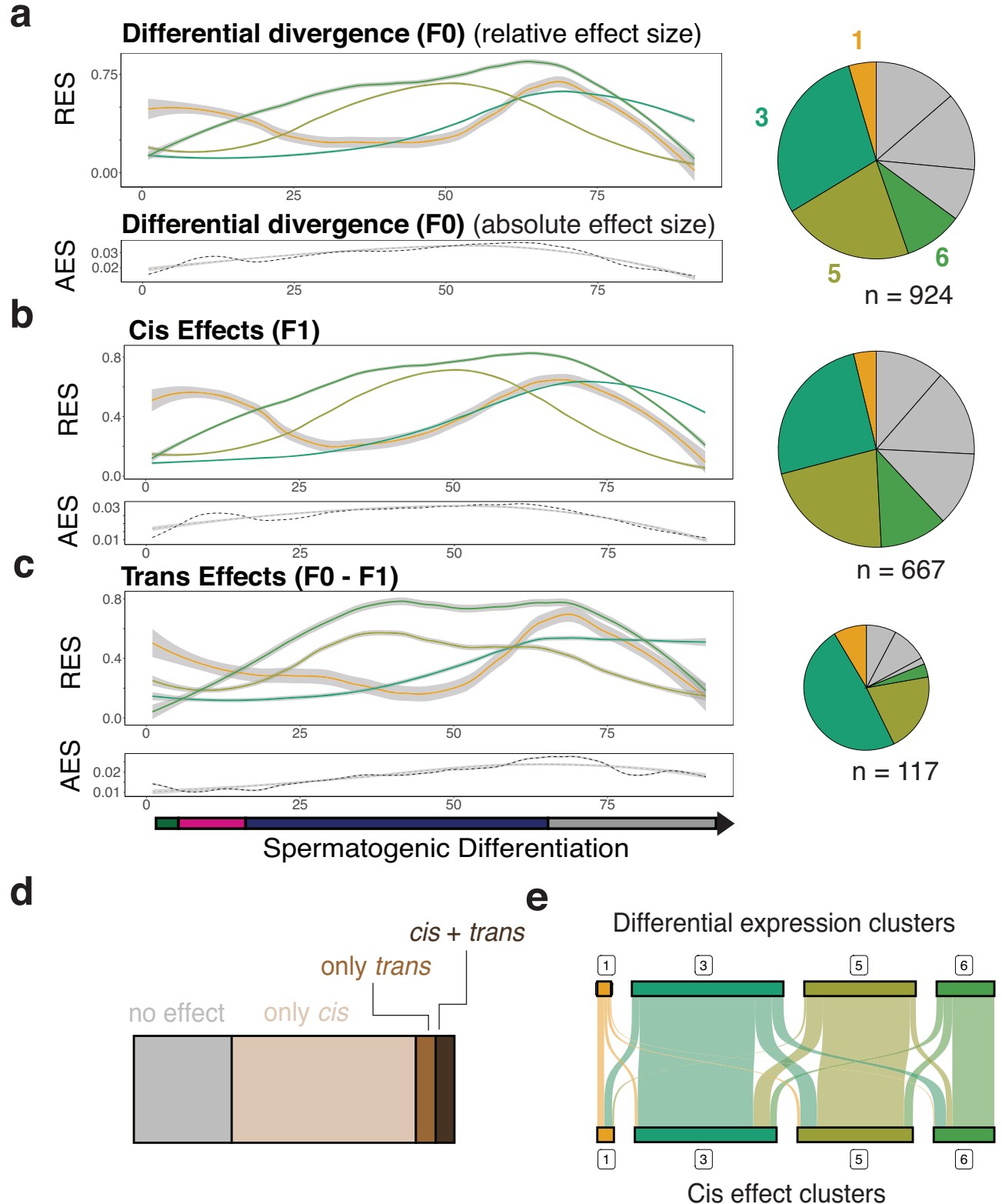

**Fig. 5 | Dynamic expression divergence between mouse sub-species is predominantly caused by differentiation-dependent *cis*-effects. a** Hierarchical clustering of the temporal profile of transcriptional divergence, identifying 7 distinct clusters. Left: Average of F0 allelic imbalance (expression divergence) across differentiation for four selected clusters where allelic imbalance peaks late in differentiation (see Fig. S11a,b for results from all remaining clusters). Allelic effects are scaled to relative values between zero and one; grouped by hierarchical clustering. Right: Proportion of genes assigned to each cluster. Bottom: Average effect size for all genes with dynamic differential expression across differentiation irrespective of cluster membership. n specifies the number of genes. Trajectories are derived by LOESS-smoothing, shown with 95% confidence intervals. **b** As (**a**), for F1 allelic imbalance, i.e. *cis*-effects. **c** As (**a**), for *trans*-effects. Bottom panel shows positions of cells in different cell types across differentiation. **d** Proportion of genes with evidence for 1) no dynamic genetic effect, 2) for dynamic *cis*, 3) dynamic *trans*, or 4) both types of effects among all genes showing dynamic differential expression between parental strains. (**e**) Overlap of genes among clusters of dynamic *cis* and dynamic differential expression.

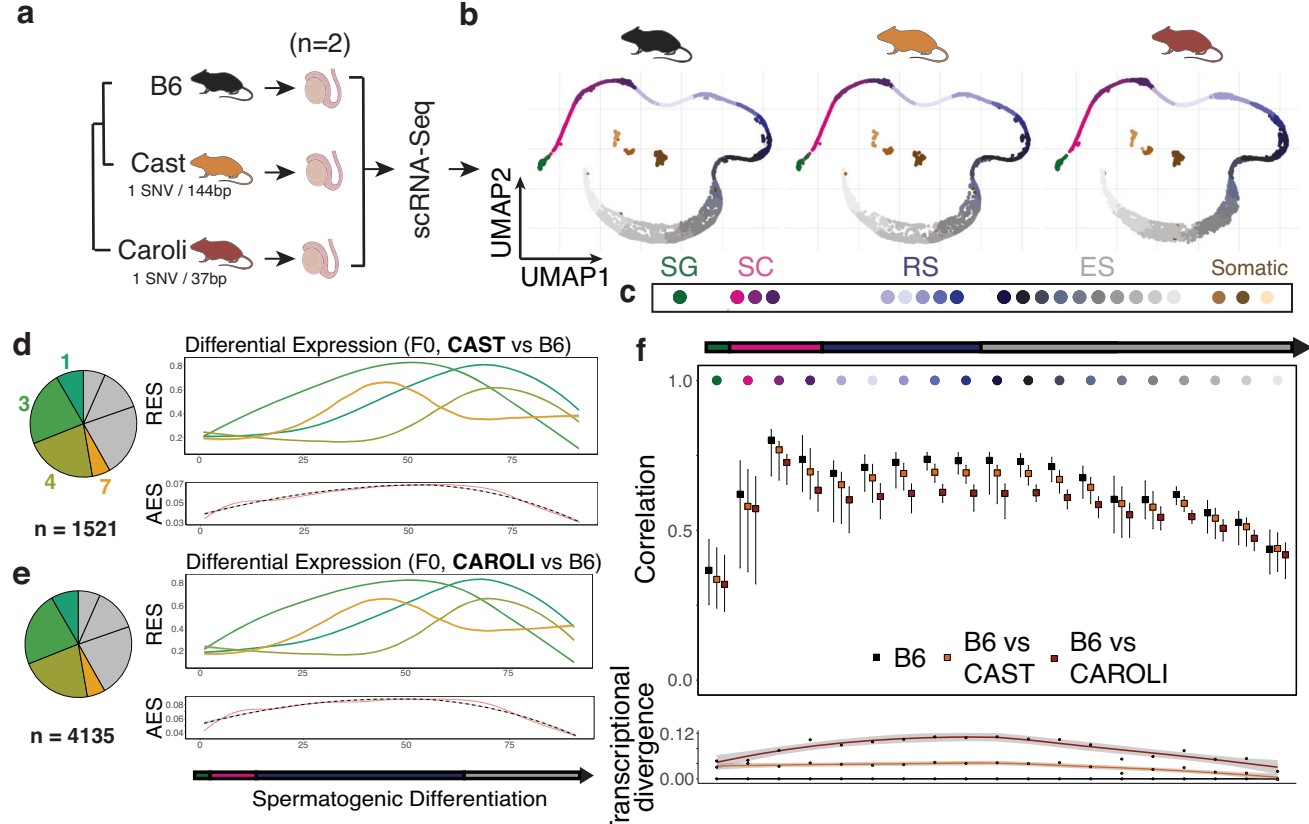

**Fig. 6 | Within-species allelic imbalance corresponds to between-species transcriptional divergence. a** Testes from C57BL/6, CAST/EiJ and CAROLI/EiJ mice were subjected to 10x Genomics scRNA-Seq in duplicates. The mouse and testis icons were created with Biorender.com. **b** Sperm differentiation was highly conserved among all three mice, with similar differentiation trajectories and cell type distributions. The UMAPs display 6098 (B6), 5080 (CAST) and 11,380 (CAROLI) cells respectively. **c** Legend showing the representation of sub cell types by colours. Hierarchical clustering of dynamic differential expression trajectories for CAST / B6 and CAROLI / B6 (**d** and **e**, respectively). Left: Proportion of genes assigned to each cluster. Right: Average relative expression divergence across pseudotime for genes assigned to the four clusters peaking late in differentiation. Bottom: Average effect size for all genes with dynamic differential expression across differentiation irrespective of cluster membership. n specifies the number of genes. Trajectories are derived by LOESS-smoothing, shown with 95% confidence intervals. **f** Correlation-based measure of expression divergence. Shown are the average correlation coefficients between pairs of B6 versus B6 cells, B6 vs CAST and B6 vs CAROLI cells. Whiskers represent top- and bottom 10% quantiles. Bottom: Difference between average B6-B6 and B6-CAST/B6-CAROLI correlations. The smoothing line is estimated by loess-regression and shown with 95% confidence intervals.

To confirm these results, we also used a global correlation-based measure of transcriptional divergence, similar to approaches commonly employed for molecular evolution analyses[17,19,35]. Within each cell type, we computed cell-cell correlation coefficients within B6 as a reference, and then between species (Methods). We define transcriptional divergence as the median difference between these correlation coefficients (Fig. 6f). This analysis again indicates round spermatids as having the strongest divergence.

Taken together, our results show that the concentration of *cis*- and *trans*-effects in round spermatids identified within species is mirrored by relatively higher transcriptional divergence observed in round spermatids between species.

## Discussion

To understand how *cis*- and *trans*-acting genetic effects dynamically impact gene expression and transcriptional divergence in vivo, we analysed spermatogenesis as a model differentiation process. Using single-cell transcriptomics in F1 mice, we revealed that developmental transitions are associated with context-dependent regulatory variation. Our experimental strategy further enabled us to connect within-species genetic effects to between-species gene expression differences, demonstrating how *cis*- and *trans*-effects jointly contribute to the increased transcriptional divergence found in round spermatids.

Our study is the first demonstration of context-dependent *cis*- and *trans*-acting genetic regulation across a continuous path of mammalian differentiation in vivo at single-cell resolution. We discovered that cell type-dependent *cis*-regulatory variation is surprisingly pervasive, representing at least 44% of *cis*-effects, and that these genes are under reduced evolutionary constraint. Indeed, this balance between persistent and dynamic *cis*-effects closely mirrors those active in *c. elegans* embryos[2]. While our study does not identify specific interactions between individual variants and target genes these could be fine-mapped using a larger number of F2 crosses[36,37] to fully elucidate the genetic mechanisms shaping species-specific expression. Compared to *cis*-effects, *trans*-effects occur less prevalently but exhibit dynamic regulatory effects at a similar rate than *cis*-effects.

Our study affords two major insights into the genetic component of transcriptional evolution. First, while prior studies have implicated *cis*-effects with transcriptional divergence in single tissues[14,25,30], we here provide evidence that genetic changes between species can arise from context- and cell-type specific *cis*-acting regulatory effects. Second, these dynamic genetic effects are strongest in round spermatids, leading to their increased transcriptional divergence. In addition to pervasive transcription of genes under low constraint as suggested previously[19–21], transcription in round spermatids shows increased sensitivity to regulatory variation. Both insights were uniquely possible

because of the high conservation of the murine genomes and cell types, which allows direct alignment of different species on the level of genes and across differentiation stages. The insights into regulatory architecture we gained are likely pervasive in mammalian differentiation and development, as all cell state changes feature substantial remodelling of gene regulatory networks, including local, *cis*-acting TF binding as well as up- and down-regulation of *trans*-acting master regulators. As such, our study in a genetically well defined system complements related work that profiles highly divergent species[19,20,38].

Single-cell resolution can capture the temporal resolution required to study dynamic genetic effects, which are impossible to resolve from analyses based purely on individually isolated cell types. Our results demonstrate that, given sufficient genetic variation, current droplet-based single-cell approaches provide sufficient sensitivity for identifying dynamic genetic effects, even for subtle *trans*-effects. However, the detection sensitivity of current technologies is lower than bulk RNA-seq and is focused on the 3′ end of genes, thus limiting our analyses to more robustly expressed genes containing allelically resolvable genetic variants. Nevertheless, our study demonstrates a powerful and easily applied strategy to disentangle context-specific genetic contributions to transcriptional divergence in embryonic development and homeostatic adult tissues. Finally, our results demonstrate that in cellular differentiation in vivo, allelic imbalance in gene expression is associated with allelic imbalance in chromatin accessibility, as shown in other contexts[39,40], suggesting that gene regulatory mechanisms are likely at least partially responsible for differential allelic usage. Novel single-cell approaches that map chromatin states, but also to directly assess mRNA synthesis and stability offer exciting opportunities to further dissect these dependencies and how they relate to cell state[41].

Collectively, our results provide a comprehensive quantification of cell type-specific genetic effects during spermatogenic differentiation, and we elucidate how the accumulation of dynamic *cis*-effects is a major mechanism underlying cell type-specific transcriptional divergence.

## Methods
### Mouse materials
All mice were bred in-house (the C57BL/6-Ly5.1, CAST/EiJ and CAROLI/EiJ colonies were established from founders obtained from the Jackson Laboratories, Strains #002014 and #000928, #000926) in the animal facilities of the DKFZ under specific pathogen-free conditions or in the Biological Resources Unit (BRU) in the Cancer Research UK – Cambridge Institute under Home Office Licence PPL 70/7535. Mice were kept in individually ventilated cages at 24°, a humidity of 80% with fixed day/night cycles of 12 h. Mice were euthanized by cervical dislocation and all animal procedures were performed according to protocols approved by the Regierungspräsidium Karlsruhe. This investigation was approved by the Animal Welfare and Ethics Review Board and followed the Cambridge Institute guidelines for the use of animals in experimental studies under Home Office licences PPL 70/7535 until February 2018 and PPL P9855D13B from March 2018. All animal experimentation was carried out in accordance with the Animals (Scientific Procedures) Act 1986 (United Kingdom) and conformed to the Animal Research: Reporting of In Vivo Experiments (ARRIVE) guidelines developed by the National Centre for the Replacement, Refinement and Reduction of Animals in research (NC3Rs). F1 hybrid mice were generated by crossing C57BL/6-Ly5.1 with CAST/EiJ mice. All mice were sacrificed after 8 weeks of age, when spermatogenesis is fully established.

### 10x Genomics scRNA-Seq
scRNA-Seq of murine testicular tissue using the 10x Genomics platform was performed similarly to Ernst et al., 2019[18]. For the cross-species dataset, single-cell suspensions were generated by enzymatic digestion using 25 mg/ml Collagenase A (Sigma, 10103578001), 25 mg/ml Dispase II (Sigma, D4693) and 2.5 mg/ml DNAse I (Sigma, 10104159001) I for 30 min at 37 C. For each sample, 10,000 cells were loaded into one channel of the Chromium™ Single Cell A Chip (10X Genomics ®, 1000009) and scRNA-Seq libraries were generated using the Chromium™ Single Cell 3′ Library & Gel Bead Kit v2 (10X Genomics ®, 120237) according to the manufacturer's instructions. Illumina short-read paired-end sequencing was performed using a HiSeq2500 with read lengths 26 bp on read 1 and 98 bp on read 2. Note that the two B6 libraries in this dataset are identical to samples published in Ernst2019.

For the F1 dataset, digestion was performed using 5 mg/ml Collagenase A, 5 mg/ml Dispase II and 2.5 mg/ml DNAse I, libraries were generated using the Chromium™ Single Cell B Chip (10X Genomics ® 1000073) and Single Cell 3′ Library & Gel Bead Kit v3 (10X Genomics ®, 1000075) and libraries were sequenced on a NovaSeq 6k with 28 bp read 1 and 94 bp read 2. The six replicates were generated in three experimental batches, each comprising two individuals from both parental strains and two F1s. Further information about the sequenced individuals can be found in Supplementary Data 1.

### ATAC-Seq of F1 spermatocytes
We isolated spermatocytes based on nuclear DNA content using fluorescence-activated cell sorting as described previously with modifications[18]. After preparation of single cell suspensions from testes, cells were stained with 5mug/mul Hoechst 33342 (R37165, ThermoFisher) for 45 min at 37 °C. Cells were resuspended in phosphate buffered saline (PBS, Sigma) with 1% Foetal Calf Serum (FCS, Gibco, 16140071) with propidium iodide (P4170) at a final concentration of 1 µg/ml, and 50.000 cells were sorted on a BD FACSAria Fusion machine (Hoechst: Excitation 405 nm, 450/50 filter, PI: Excitation 488 nM, filter 616/23). We then performed bulk ATAC-Seq as described with modifications[42]. Cells were washed with 500 µl ice-cold PBS and incubated for 3 min in 50 µl cell lysis buffer (10 mM Tris-HCl pH 7.5 (AM9850G, LIFE Technologies), 10 mM NaCl (AM9760G, Thermo-Fisher), 3 mM MgCl (AM9530G, ThermoFisher), 0.1% NP-40 (85124, LIFE Technologies), 0.1% Tween-20 (P1379, Sigma), 0.01% Digitonin (BN2006, LIFE Technologies)) on ice. 1 ml wash buffer (10 mM Tris-HCl pH 7.5, 10 mM NaCl, 3 mM MgCl, 0.1% Tween-20) was added, cells were centrifuged for 10 min at 500 g and 4 °C. Transposition of nuclei was performed by adding 50mul transposition mix (25mul 2x TD buffer (Illumina, 20034197), 16.5mul PBS, 0.5mul 10% Tween-20, 0.5mul 1% Digitonin, 2.5mul tagment DNA enzyme (Illumina, 20034197)) to the pellet and incubation at 37 °C for 30 min. After incubation, DNA was isolated using MinElute PCR Purification Kit (28006, Qiagen) and libraries were PCR-amplified using NEBNext® High-Fidelity 2X PCR Master Mix (M0541S, NEB) with appropriate primers. Libraries were sequenced on an Illumina NextSeq2000 machine.

### Expression quantification of 10x scRNA-Seq data for different species
Genomic references for C57BL/6 (GRCm38), CAST/EiJ and CAROLI/EiJ were generated using CellRanger *mkref* (v3.1) using sequence and gene annotations from ensembl (release 94). Filtered count matrices were generated using CellRanger *count* (v3.1) using default settings. For the F1 dataset, a joint reference was constructed based on the GRCm38 reference where SNP positions between mm10 and CAST/EiJ were N-masked. These SNPs were derived from (Keane2011, ftp://ftp-mouse.sanger.ac.uk/current_snps/mgp.v5.merged.snps_all.dbSNP142.vcf.gz). Total expression quantification using CellRanger count (v3.1) was then performed against this reference for C57BL/6, CAST/EiJ and hybrid mice.

### Low-level analysis of scRNA-Seq data and cell type annotation
Low-level analysis of scRNA-Seq data was performed similarly to Ernst et al., 2019 and largely using functions from the *scran* (v1.20.1) and

*scater* (v1.20.1) R packages[43,44]. First, cells with less than 500 UMIs and 500 detected genes were removed. Next, counts were normalised using the *computeSumFactors* function and log-transformed. For cell type annotation, we used mutual nearest neighbour-based batch correction using the function *MNNcorrect* with the library as batch variable to exclude species-specific and technical variation across samples[45]. The resulting corrected matrix was used for dimensionality reduction by principal component analysis (*prcomp, stats*, v4.4.0), tSNE (*Rtsne, Rtsne*, v0.15) and UMAP (*umap, umap*, v0.2.7.0). To identify cell type clusters, we used graph-based community detection using the Louvain algorithm implemented by the functions *buildSNNGraph* and *cluster_louvain* of the package *igraph* (v1.2.10). Cell type labels were defined as in Ernst2019: For the evolutionary dataset, cell type labels were available from Ernst2019 for the B6 libraries, which could be used to annotate cluster identities. For the F1 dataset, equivalent clusters were defined using marker genes for somatic cells (Sertoli, Leydig and Immune / other structural cells, which were discarded for most further analysis) and the established order of cell types during spermatogenic differentiation (Figs. S1a–g, S12). A continuous pseudo-temporal ordering through germ cells was derived using principal curve fitting using the package *princurve* (v2.1.6), based on the first two principal components fitted across all cells. For the cross-species dataset, a pseudotime ordering across cells was derived by computing the median pseudotime for the 50 nearest neighbours in F1 dataset (Fig. S12).

## Allele-specific quantification

To quantify allele-specific expression, we first annotated the output bam file from cellranger (possorted.bam) with the B6/CAST[46] SNPs using a modified script from the WASP-pipeline (find_intersecting_snps_10x.py). We then counted individual reads if they contained one or more alleles from the maternal or paternal haplotype, while discarding UMI duplicates and reads overlapping indels. We discarded reads with conflicting SNP identities (<0.1% of all reads) as likely sequencing errors. To compare allelic data between F1 mice and F0 parents, we quantified all libraries against the N-masked reference and only considered reads that overlapped segregating SNPs. This approach also validated that >98% of segregating reads from F0 animals were assignable to the correct reference allele (Fig. S1k). To obtain similar sequencing depth per allele for the F0 and F1 mice, we downsampled the F0 libraries to 50% of reads, as at equal sequencing depth per library, the coverage of each allele in the F1 samples will be half of the F0 samples (Fig. S1k). We further discarded 71 mitochondrial and X-chromosomal genes which only showed reads mapping to the reference (maternal) allele and 7 genes with a strong paternal bias in the F1 but no bias in the F0 as likely mapping errors. We found that around 25.82% of reads were assignable to either the maternal or the paternal haplotype across samples. Depending on the application, we quantify allele-specific expression either as an allelic ratio B6 / (B6 + CAST) or as a (log2) allelic fold-change, log2 (B6 / CAST).

## Categorization into regulatory categories for discrete cell types

Genes were categorised by regulatory mechanisms as in Goncalves et al., 2012[14], considering genes with > 100 allele-specific reads per sample. We defined statistical models based on negative binomial (for the parental strains) and beta-binomial distributions (for allelic data from F1) for each gene (Supplementary Methods). The overdispersion parameters for the negative binomial distributions were computed using the function *estimateDisp* from the *DESeq2* package. The regulatory categories (*conserved, cis, trans, cis + trans*) were defined by constraining parameters in each model and fitted using maximum likelihood estimation in the function *mle (stats4 v4.2.2)*. We then used the Bayesian information criterion (BIC) to assign the most likely regulatory category and to quantify the strength of evidence for the data under each model against the conserved model (Fig S1m,n). As a more stringent classification of genetic effects, we only considered genes with a difference in BIC of at least 4 when compared to the conserved model. As a comparison, we also computed differentially expressed genes between the F0 strains using the *DESeq* function in the *DESeq2 (v1.32.0)* package.

## Detection of differential allelic imbalance between cell types using gLMs

To detect differential allelic imbalances between cell types, we employed generalised linear models with a binomial likelihood as implemented in the package *VGAM* (v1.1.5). We fit a full model using the formula intercept + cell type + library, and compared it to a reduced model without the cell type component. Significance was evaluated using likelihood-ratio tests and resulting *p*-values were adjusted for multiple testing using the Holm-Bonferroni correction.

## Dimensionality reduction based on allelic imbalance

We sought to disentangle allele-specificity from levels of gene expression, in order to perform dimensionality reduction to capture cell type-specific structure of allelic imbalance. To this end, we defined a score that quantifies the confidence of the presence of allelic imbalance per gene and per cell as the log-likelihood-ratio of observing a given pair of alternative and reference counts given the observed ratio and the underlying probability 0.5. For a given gene and cell, scores were defined as 0 if no reads were observed. The resulting score matrix was subjected to principal component analysis and UMAP (*umap*, v0.2.7.0).

## Detection of dynamic allelic imbalance (*cis*-effects) using scDALI

We detected allelic imbalance using the scDALI-framework[31]. To test for allelic imbalance independent of cell type, *p*-values were derived using a likelihood ratio test against a null model with a fixed allelic imbalance of 0.5 (function *BetaBinomLRT*). Dynamic allelic imbalance was determined using the heterogeneous scDALI score test with a cell state kernel defined by a degree 3 polynomial on the pseudotime. P-values were adjusted for multiple testing using the Holm-Bonferroni procedure and considered significant with an adjusted p-value < 0.01. We then defined genes with persistent allelic imbalance as those that did not show dynamic effects but were significant in the likelihood ratio test.

## Clustering of allelic trajectories

Using scDALI, we derived latent allelic trajectories for each gene using gaussian process regression with an RBF-kernel for all genes with significant dynamic components. We next scaled these trajectories by subtracting the mean and dividing by the standard deviation (z-scoring). We then performed hierarchical clustering (*hclust* from the R package stats) on the scaled trajectories and identified 7 clusters based on a sum of squared distance metric. We also derived the same number of clusters for total log-transformed and z-scored gene expression measurements, considering both reads with and without allelic assignment (Fig S4).

## Analysis of allele-specific ATAC-Seq data

ATAC-Seq reads were trimmed using trimmomatic (v0.38)[47] and mapped to the mm10 genome build using bowtie2 (v2.3.5.1)[48]. Peaks were called using macs2 (v2.1.2.1) with the --nomodel --extsize 200 --shift --100 --call-summits parameters. We finally used the find_intersecting_snps.py script from the WASP package[46] (v0.3.4) to annotate reads with SNVs between the B6 and CAST genomes and used a modified version of the count_allelic.py script from scDALI ([https://github.com/tohein/scai_utils](https://github.com/tohein/scai_utils)) to count allele-specific reads within each peak which were used for further analysis. We found a consensus peak set between our two replicates using the *mergeByOverlaps* function (GenomicRanges, v1.44.0, with the argument minoverlap = 0.9) and

quantified its genomic distribution using the *annotatePeak* from the ChIPseeker package[49] (v1.28.3). We annotated each peak with its closest gene, excluding peaks for which the closest gene was >20 kb away, only retained peaks with at least an average of 50 allele-specific reads and quantified allelic imbalance (AI) in chromatin accessibility as the average read count ratio B6 / (B6 + CAST) across both replicates. We then performed an over-representation analysis of open chromatin sites with allelic imbalance at genes with allelic imbalance in gene expression. To this end, we quantified the effect size of AI for a gene or peak as $d = |AI - 0.5|$ and computed the fraction of genes with associated ATAC-Seq peak with $d > 0.1$. We obtained a random distribution by shuffling the observed AI estimates randomly across peaks. Finally, to investigate the association between dynamic AI and chromatin accessibility, we considered the set of genes with dynamic AI, ranked it by the differential in AI between spermatocytes and spermatids and quantified the strength of AI in associated ATAC-Seq peaks.

### Analysis of allelic imbalance across up- and down-regulation in gene expression trajectories

To compare allelic imbalance during the dynamics of gene up- and down-regulation, we first identified genes that showed both up- and down-regulation during spermatogenesis based on total read counts. To this end, we fitted loess-regression curves between pseudotime and log total read counts of each gene using the *loess* function (stats) across the interval in which a gene was expressed, identified the point of maximum expression and retained all genes that did not peak in the first or last interval of pseudotime (100 evenly spaced intervals). We then split the genes expressed trajectory into 5 quantiles and compute allelic imbalance across all cells within each quantile, which is then visualised.

### Detection of dynamic trans-effects

To detect dynamic *trans*-effects, we first require a joint coordinate system in which expression values between F0 and F1 strains can be compared. To this end, we used a joint coordinate system for all samples (see **Low-level analysis of scRNA-Seq data**) to derive a pseudo-temporal ordering for all cells from all samples and verified that the resulting trajectory was covered similarly in all samples (Fig S4). We next split cells into 100 pseudotime intervals and computed allelic ratios for all genes between the B6 and CAST alleles (between F0 B6 and CAST mice and between alleles within F1 mice). As the F0 allelic quantifications are derived from different mice and not necessarily sequenced to the same depth, we normalise bin-wise expression estimates per allele across all genes before computing allelic ratios. Next, we modelled allelic trajectories using a generalisation of the Gaussian process model underlying scDALI. The two trajectories are modelled as a realisation from a co-regionalised Gaussian process with a kernel $\mathbf{B} \otimes \mathbf{K}$, where K is a kernel matrix and the off-diagonal elements in $\mathbf{B}$ represent the covariance between F0 and F1 trajectories. Furthermore, constant shifts between F0 and F1 can be encoded by varying mean functions. Based on this, we define trajectories with no or only *cis*-effects by fixing mean functions and $\mathbf{B}$. Persistent trans-effects allow for varying means and dynamic *trans*-effects allow for a full-rank $\mathbf{B}$. We are using a Matern kernel for K and all hyperparameters are fit using variational approximations in the *GPflow* (v2.1.4) python package[50]. We then use the evidence lower bound (ELBO) of each model as an approximation to the marginal likelihood of the data under the respective model and derive Bayes Factors for each gene which quantify the model evidence for persistent and dynamic effects. To compare *cis*- and *trans*-effects, we also define a model with a constant kernel against which dynamic *cis*-effects can be detected. We consider model evidence sufficient when the log Bayes Factor to a given null model exceeds 10. For further details of the modelling approach, refer to the Supplementary Methods.

### Detection of dynamic differential expression

Analogously to the detection of dynamic *cis*-effects using GP regression, we nominate genes with dynamic differential expression, considering all genes with at at least 1000 reads across samples. To this end, we use gaussian process regression on the bin-wise allelic ratios of F0 mice and derive Bayes Factors comparing a model with a dynamic to a constant kernel.

### Joint clustering of dynamic *cis*, *trans* and differential expression effects

To derive joint patterns of total expression, genetic effects and differential expression trajectories, we first computed bin-wise estimates of *cis*-effects ($|AI - 0.5|$ in F1), differential expression (abs(allelic ratio - 0.5)) and *trans*-effects ($|AI\_F1 - AI\_F0|$). This was done for all genes with detected dynamic effects (log BF > 10). We then smoothed these trajectories by taking the average across 5 bins and computed the average across all genes to compute the total dynamic effect estimate. We then scaled each individual genes to the 0 - 1 range and subjected them jointly to hierarchical clustering (potentially including *cis*, *trans* or *de* effects for the same gene). We then computed average cluster trajectories per effects group (*cis, trans, de*) and cluster.

### Correlation analysis

For the cross-species dataset, we quantified transcriptional divergence between species for individual cell types. For a given cell type, we computed Spearman correlations of log-transformed expression values across genes for each pair of B6 cells. We then computed the same distribution of correlation coefficients between pairs of B6 and CAST / CAROLI cells. We then defined transcriptional divergence as the median difference between the B6 correlations and the B6-CAST / B6-CAROLI correlations.

### Reporting summary

Further information on research design is available in the Nature Portfolio Reporting Summary linked to this article.

## Data availability

All newly generated sequencing data has been deposited in ArrayExpress under the accession number E-MTAB-11602. The B6 samples of the cross-species comparison are deposited in ArrayExpress under the accession number E-MTAB-6934. The spermatocyte ATAC-Seq has been deposited under the accession number E-MTAB-12685. All other relevant data supporting the key findings of this study are available within the article and its Supplementary Information files or from the corresponding author upon request. A reporting summary for this Article is available as a Supplementary Information file. Genomic files and annotations are available from ensembl http://www.ensembl.org/Mus_musculus/Info/Index. Variants between B6 and CAST mouse strains are available at https://ftp.ebi.ac.uk/pub/databases/mousegenomes/REL-1505-SNPs_Indels/mgp.v5.merged.snps_all.dbSNP142.vcf.gz.

## Code availability

All code to reproduce the results presented in this paper can be found under https://github.com/PMBio/ase_spermatogenesis/.

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

## Acknowledgements

The authors thank members of the Stegle and Odom lab for discussions, Meike Schopp and Marie-Luise Koch for management of mouse colonies, Fritjof Lammers for support with data management and Paul Ginno for assistance with flow sorting. We further thank the DKFZ animal facility and the DKFZ genomics and proteomics core facility for their assistance. This work was supported by core funding from the German Cancer Research Center (O.S & D.T.O) and the European Molecular Biology Laboratory (O.S & J.C.M), Cancer Research UK (D.T.O & J.C.M), the European Research Council (grant agreement IDs 810296 / DECODE to O.S. and 788937 / CTCFStableGenome and 615584 / EvoGeneticsTFBinding to D.T.O), Wellcome Investigator Award (202878_Z_16_Z to D.T.O.), as well as the Bundesministerium für Bildung und Forschung Germany, project MERGE, Förderkennzeichen 031L0174C.

## Author contributions

J.P. generated F1 single-cell RNA-Sequencing data, analysed all data and generated all figures. T.H. developed scDALI and contributed to its application. C.E. generated cross-species single-cell RNA-Sequencing data under supervision from J.C.M. N.E. performed preliminary analysis of the cross-species dataset and generated code for preprocessing of single-cell RNA-Sequencing data under supervision from J.C.M. R.E.W. contributed to the generation of functional genomics data. M.S. contributed to generation of single-cell RNA-Sequencing data. J.P, O.S and D.T.O interpreted results and wrote the manuscript with input from all authors. O.S. and D.T.O supervised the study.

## Funding

## Competing interests

J.C.M has been an employee of Genentech since September 2022. The other authors declare no competing interests.
