## [Peer Review File · Nature Communications]

The dynamic genetic determinants of increased transcriptional divergence in spermatidsEditorial Note: This manuscript has been previously reviewed at another journal that is not operating a transparent peer review scheme. This document only contains reviewer comments and rebuttal letters for versions considered at *Nature Communications*.

REVIEWER COMMENTS

Reviewer #1 (Remarks to the Author):

Thank you for revising the manuscript, performing additional experiments, and considering the conclusions drawn. Collectively, these revisions address all the comments and suggestions by the three reviewers. Thank you.

Reviewer #2 (Remarks to the Author):

Much of the same holds true from the previous report. “The study is technically sound and the dataset will be a useful genomic resource to study gene expression changes in F1 individuals during spermatogenesis. However, the manuscript lacks a clear message, is hard to follow, does not consider all aspects of spermatogenesis gene regulation, and there are concerning overstatements” Below are some examples of these concerns:

1. In the title what is “accelerated spermatid evolution”? Is gene expression, gene sequence, cellular morphology, etc. The title makes it sound like there will be comparison of spermatid biology across a range of species, however that is not the case. The title should be revised to echo the findings of the manuscript. The authors are likely arguing for accelerated transcriptional evolution, but the manuscript lacks convincing data that the transcriptional regulation is evolutionarily accelerated versus relaxed constraints on transcription in spermatids. Could the authors change the title to “increased transcriptional divergence in spermatids” instead of accelerated evolution. Along the same lines the manuscript, which is largely evolutionarily focused should be guided by the basic question of whether the transcriptional changes are under selection or under relaxed constraint. Can the authors support either reduced constraint or positive selection with their data? Premeiotic and Meiotic germ cells are under stronger evolutionary constraints, because both males and females share similar cell types. However, post-meiotic cells, which are spermatids, do not exist in females, so the increased transcriptional divergence the authors observed in post-meiotic cells may be due to it being a male-specific process, unencumbered by selection acting to preserve gene expression patterns between males and females.

2. The manuscript still lacks a clear biological question laid out in the introduction. The authors state (lines: 97-98) “How changes in DNA sequence impact molecular traits that ultimately drive phenotypic and disease remains poorly understood”, but this is not addressed in the manuscript. The experiments are assessing the relationship between genetic and phenotypic variation across evolutionary timescales, which is different from disease associated variation.

3. There should be a clear definition of a cis- vs trans-effect in the manuscript. Ideally as a figure and how the effects of cis- and trans-regulation are assessed. The lack of this fundamental concept will make the entire manuscript difficult to read, interpret and evaluate for any reader, including experienced geneticists. Additionally, the term “regulatory mechanism” is also poorly defined and perhaps misleading based on their findings.

4. The language and terminology throughout the manuscript lack clarity. The manuscript would benefit from more concise, specific language. Here are a few (of MANY) examples:

a. Lines 55-56: What is a “regulated gene” versus a “gene” are not all genes regulated? Do the authors mean “expressed genes”?

b. Lines 129-131: “Our data reveal pervasive cis- and trans-effects with either persistent effects across differentiation or dynamic allelic regulation across the cell types in the differentiation trajectory”

Can this be more simply written as “pervasive cis- and trans- effects are either persistent throughout differentiation or cell-type specific?”

c. Lines 190-195: This identified substantial differential cis- regulatory effects (differences in allelic fold changes) between cell types, for example 411 genes between spermatocytes and round spermatids (FDR < 10%, absolute difference in log₂ aFC between the two cell types > 0.5) and spermatocytes and elongating spermatids (411 genes, FDR < 10%, absolute difference in log₂ aFC > 0.5) (Fig 1e, f).

Does this mean there are 411 genes with allele-specific expression patterns in the different cell types?

d. The word “dynamic” is over-used throughout the paper in several different contexts. The authors should identify every instance where they use the term “dynamic” and replace it with more specific terminology. For example, in line 238: Instead of using the term “dynamic” to describe allelic differences within cell types, can the authors use “cell-type specific”

5. Lines 177-178: “Reassuringly, these numbers are similar to the total number of genes identified using conventional differential expression analysis between the founder strains (Fig S6c) “ The number of genes is similar but is it a similar gene set?
6. Lines 300-303: “Furthermore, we observed that allele-specific expression close to genes with asCA (in spermatocytes) were strongest in spermatocytes, suggesting that dynamic cis-effects can be driven by cell type-specific changes in chromatin accessibility (Fig S7e,f) “How can the association be strongest in spermatocytes if you only looked in spermatocytes?
7. The ATAC-Seq findings are problematic for the overall findings of the paper. It seems the authors are claiming that sequence variation drives allele-specific expression, but the ATAC-Seq findings would suggest that it is chromatin based. Are the authors arguing for one or the other or a combination of both influencing the gene expression divergence?
8. Lines 567-571: “we here provide evidence that a substantial fraction of fixed genetic changes between species likely arose from highly dynamic and cell- type specific cis-acting regulatory effects. Second, our results provide direct evidence of increased fixation of regulatory changes in later spermatogenesis, which has been proposed previously in other vertebrates”? What does “increased fixation of regulatory changes” mean? How do the authors know they are fixed, as they may be just as variable within *Mus musculus* and *Mus castaneus* populations as they are between the two strains.

Reviewer #4 (Remarks to the Author):

Reviewer report for “The dynamic genetic determinants of accelerated spermatid evolution” by Panten et. al. 2023

In this article, the authors have used single-cell RNA and bulk-ATAC sequencing technology to examine gene expression and chromatin accessibility changes occurring during spermatogenesis. They compare gene expression in F1 males to males from parental (F0) strain background. They then go onto classify regulatory mechanisms, i.e. cis- versus trans- acting effects. A key part of the work that follows is based around this classification. They describe a large number of transcriptional changes that occurs during spermatogenesis, and ascribe this to both cis- and trans- effects on gene regulation. They state that cis-effects were generally more pronounced than trans- effects overall, and this also seen in cell sub-types. They detect increased transcriptional divergence in round spermatids stages, and from modelling suggest that the basis for this is due to allelic imbalances that exist within species.

In essence I believe the study is motivated to address how differences in gene expression levels between genetically distinct individuals, is driven by these differences in cis- and in trans-. Leveraging on single-cell RNA-seq performed on spermatogenic cells (which have a unidirectional differentiation dynamic), the authors reconstruct a trajectory of differentiation and further make comparisons between F1 males and genetically distinct parental mouse lines to ascribe regulatory effects.

Over the course of revision, the authors have including new supporting ATAC-seq data / analysis, and have revised the manuscript which supports their findings. The study is technically solid, and experiments and bioinformatic analysis have been performed to a high standard. The datasets generated will also be very useful to a number of research areas (e.g. genomics and genetics more generally). Overall the article is worthy to be considered for publication in Nat Comms.

Main suggestions:

1. I note that the manuscript has already been reviewed, that significant feedback from referees was already provided, and I would be broad agreement with the comments. My view is that the manuscript is generally written well, however it is pitched at a very high level. The scope of the work is vast and the subject area is complex. Nevertheless, I did find the text and some figures difficult to follow and there was also the lack of a simple message in the sub-sections. I found myself re-reading sections in order to understand. I believe that it may also be helpful to reorder the supplementary figures to how they appear in the text to make them easier to refer to for the general readership.

2. Inclusion of a html notebook and / or relevant code of how the analysis were performed be helpful for better understanding reproducibility.

3. Reviewer #3 comment: "It's also curious though that the authors identify many cis-trans genetic elements in spermatids although these cells are transcriptionally quiescent. Do these mutations affect somehow RNA stability since these populations are transcriptionally

quiescent?”

- it is important that the authors explain this better in the text, as a major thrust of the paper centres around accelerated spermatid evolution, perhaps a sub-section under “Dynamic changes in cis-acting genetic effects on transcription across sperm differentiation”. NB - The author revision note refers to a revised Fig.2d which I could not locate.

4. I could not work-out how Fig.1g was generated, nor what it is trying to convey.

- relatedly, from the RNA-seq point of view, it would be helpful to expand or highlight a few more genes in Supp. Fig 1f for the benefit reader.

5. Lines 269 – 279: I could not follow this paragraph – the sentence “The two largest clusters (cluster 1 and 2; collectively covering 45.61% of all genes) showed mirror-image changes to allelic imbalance...” appears a little misleading. The percentage stated (45.61%), at least how it appears on the heatmap, appears to be high, and an x-axis label is not shown. Does the x-axis refer to individual cells ordered in pseudotime?

5. Lines 465 - 467: Please can you point the reader to the figure that supports this sentence.

Minor:

1. Line 491: suggest removing the word “Briefly”, as it makes it sound like the methods section.

Please find our responses to the Reviewer's comments in this file. We have attempted to structure our response by directly commenting on individual points raised and systematically referring to the revised manuscript (by line numbers and Figures 1-6, Figures S1-13), in which we highlight major textual changes in blue.

Key:

- Reviewers' comments
- Our response

REVIEWER COMMENTS

Reviewer #1 (Remarks to the Author):

Thank you for revising the manuscript, performing additional experiments, and considering the conclusions drawn. Collectively, these revisions address all the comments and suggestions by the three reviewers. Thank you.

We thank the reviewer for this assessment.

Reviewer #2 (Remarks to the Author):

Much of the same holds true from the previous report. “The study is technically sound and the dataset will be a useful genomic resource to study gene expression changes in F1 individuals during spermatogenesis. However, the manuscript lacks a clear message, is hard to follow, does not consider all aspects of spermatogenesis gene regulation, and there are concerning overstatements” Below are some examples of these concerns:

We thank the reviewer for valuing the quality and scope of the datasets we present. To improve clarity we have substantially revised our manuscript, more specifically emphasizing the key results. Specifically, the main insights of our manuscript is the characterization of context-dependent *cis* and *trans* gene regulatory effects that vary across cellular differentiation. While we still believe these findings have implications for evolutionary biology, we have toned down this aspect of the paper.

1. In the title what is “accelerated spermatid evolution”? Is gene expression, gene sequence, cellular morphology, etc. The title makes it sound like there will be comparison of spermatid biology across a range of species, however that is not the case. The title should be revised to echo the findings of the manuscript. The authors are likely arguing for accelerated transcriptional evolution, but the manuscript lacks convincing data that the transcriptional regulation is evolutionarily accelerated versus relaxed constraints on transcription in spermatids. Could the authors change the title to “**increased transcriptional divergence in spermatids**” instead of accelerated evolution.

We thank the reviewer for highlighting this point. We would like to stress that the major thrust of the manuscript is the quantification of genetic effects, similar to previous work with F1 hybrid systems, and here we extend this analysis to the quantification of context-dependency of genetic effects. In this case this context-dependency is given by the differentiation stage. We have implemented your suggestion to change the title and to de-emphasize the evolutionary implications. Further, we have changed the wording in the manuscript to focus on transcriptional

divergence, rather than accelerated evolution (Title and Lines 50 & 60, 123, 401-429, 530, 565-568).

Along the same lines the manuscript, which is largely evolutionarily focused should be guided by the basic question of whether the transcriptional changes are under selection or under relaxed constraint. Can the authors support either reduced constraint or positive selection with their data? Premeiotic and Meiotic germ cells are under stronger evolutionary constraints, because both males and females share similar cell types. However, post-meiotic cells, which are spermatids, do not exist in females, so the increased transcriptional divergence the authors observed in post-meiotic cells may be due to it being a male-specific process, unencumbered by selection acting to preserve gene expression patterns between males and females.

We agree that this is an important distinction where we should use the appropriate terminology. Our results do not distinguish between higher transcriptional divergence in spermatids due to accelerated evolution and/or lower constraint. Indeed, the reason for this observed increase in divergence is an area of active research and has been attributed to lower selective constraint, promiscuous expression of lowly conserved genes, and an increase in fixation of adaptive changes¹⁻⁴). Our main contribution here is that we show direct evidence that also for broadly expressed genes, the genetic effects inducing strain-specific expression are stronger in spermatids. We can therefore attribute loss of constraint to gene regulatory effects, as now outlined in the discussion (Lines 534-537). We have now implemented your suggestion and refrain from emphasizing accelerated evolution in the title and throughout (Lines 123, 530, 565-568) to avoid ambiguity.

2. The manuscript still lacks a clear biological question laid out in the introduction. The authors state (lines: 97-98) "How changes in DNA sequence impact molecular traits that ultimately drive phenotypic and disease remains poorly understood", but this is not addressed in the manuscript. The experiments are assessing the relationship between genetic and phenotypic variation across evolutionary timescales, which is different from disease associated variation.

We now re-state in the introduction that we are addressing to which extent genetic effects are cell type-specific (Lines 104-106). Using a controlled system with defined genetic variation, for the first time, we can assay the context-dependency of both *cis*- and *trans*-effects comprehensively *in vivo* and this conceptually extends to human variation and disease.

3. There should be a clear definition of a *cis*- vs *trans*-effect in the manuscript. Ideally as a figure and how the effects of *cis*- and *trans*-regulation are assessed. The lack of this fundamental concept will make the entire manuscript difficult to read, interpret and evaluate for any reader, including experienced geneticists. Additionally, the term

“regulatory mechanism” is also poorly defined and perhaps misleading based on their findings.

We added the requested figure at the beginning of the manuscript (now Figure 1), which also re-introduces the F1 hybrid system previously used to assay *cis*- and *trans*-effects, and how these relate to transcriptional divergence (Lines 134-146)⁵⁻⁷. We agree that “regulatory mechanism” in this model cannot refer to a specific interaction of a gene with a specific *cis*- or *trans*-acting factor, and is therefore potentially overstated. We have reworded these instances as “regulatory class” (Line 181-187).

4. The language and terminology throughout the manuscript lack clarity. The manuscript would benefit from more concise, specific language. Here are a few (of MANY) examples:

We have addressed the individual examples below. More generally, in consultation with the handling editor, we have again heavily revised the manuscript in an attempt to comprehensively address these concerns. Our resubmission uses blue text to highlight sections that have been overwhelmingly re-written, representing well over a third of the text.

a. Lines 55-56: What is a “regulated gene” versus a “gene” are not all genes regulated? Do the authors mean “expressed genes”?

In this instance, we refer to genes under *genetic* regulation, that is, under the impact of regulatory variation as detected between the strains. We clarified this point (Line 56).

b. Lines 129-131: “Our data reveal pervasive *cis*- and *trans*-effects with either persistent effects across differentiation or dynamic allelic regulation across the cell types in the differentiation trajectory”

Can this be more simply written as “pervasive *cis*- and *trans*- effects are either persistent throughout differentiation or cell-type specific?”

Yes, and we have revised and simplified this particular sentence (Lines 119-120).

c. Lines 190-195: This identified substantial differential *cis*- regulatory effects (differences in allelic fold changes) between cell types, for example 411 genes between spermatocytes and round spermatids (FDR < 10%, absolute difference in log₂ aFC between the two cell types > 0.5) and spermatocytes and elongating spermatids (411 genes, FDR < 10%, absolute difference in log₂ aFC > 0.5) (Fig 1e, f).

Does this mean there are 411 genes with allele-specific expression patterns in the

different cell types?

It means that there are 411 genes whose allelic ratio differs between cell types. This indicates that *cis*-regulatory variation has differential effects on their allelic expression. For example, a gene might have a log allelic fold change $\log(A1 / A2) = 0$, which indicates equal expression from both alleles in cell type 1, but $\log(A1 / A2) = 1$ in cell type 2, which indicates differential expression from both alleles due to *cis*-acting genetic effects. The text around this analysis has been revised (Lines 197-203).

d. The word “dynamic” is over-used throughout the paper in several different contexts. The authors should identify every instance where they use the term “dynamic” and replace it with more specific terminology. For example, in line 238: Instead of using the term “dynamic” to describe allelic differences within cell types, can the authors use “cell-type specific”

In the genetics literature, the term “dynamic genetic effect” has previously been used to refer to the context-specific action of regulatory variants, which includes but is not limited to cell type-specific effects⁸⁻¹⁰. We now clearly define ‘dynamic’ explicitly both in the Introduction and Section 2 (Lines 134-146, 240-245), and use it throughout the manuscript in specifically that definition - as a genetic effect that varies during differentiation. For clarity, we have eliminated all instances in which dynamic was used outside of this definition (e.g. Lines 119-123).

We note, however, that “cell type-specific” is imprecise, because it relies on the definition of *discrete* cell types; in contrast, dynamic additionally encompasses changes that occur between subsets of cells within one cell type or continuously across differentiation. We demonstrate that we identify more genetic effects when considering differentiation as a continuous process rather than a series of discrete cell types (**Fig S4d-f**).

5. Lines 177-178: “Reassuringly, these numbers are similar to the total number of genes identified using conventional differential expression analysis between the founder strains (Fig S6c) “ The number of genes is similar but is it a similar gene set?

Yes, we have confirmed that and revised the text accordingly (Lines 193-194, **Fig S3c**).

6. Lines 300-303: “Furthermore, we observed that allele-specific expression close to genes with asCA (in spermatocytes) were strongest in spermatocytes, suggesting that dynamic *cis*-effects can be driven by cell type-specific changes in chromatin accessibility (Fig S7e,f)

“How can the association be strongest in spermatocytes if you only looked in spermatocytes?”

In this analysis, we were comparing spermatocytes to multiple other cell types. In brief, we compare the allelic imbalance of gene expression in different cell types to the allelic imbalance in chromatin accessibility in spermatocytes, and (perhaps unsurprisingly), expression-level imbalances in spermatocytes co-localize with allelic imbalance in CA more often than when using expression profiles from other cell types. This suggests that there is cell type-level coordination between allelic imbalance in expression and accessibility. We agree that it would be ideal to also directly include allelic chromatin accessibility in spermatids in this analysis, but ATAC-Seq in spermatids is challenging and was not successful in our hands.

7. The ATAC-Seq findings are problematic for the overall findings of the paper. It seems the authors are claiming that sequence variation drives allele-specific expression, but the ATAC-Seq findings would suggest that it is chromatin based. Are the authors arguing for one or the other or a combination of both influencing the gene expression divergence?

Our proposed scenario is that genetic variation changes chromatin accessibility (for example, due to differences in transcription factor binding) which impacts gene regulation in an allele-specific manner, which in turn changes allele-specific expression. Indeed, it has been shown that allelic variation caused by sequence variation in chromatin accessibility and expression is correlated¹¹. Non-sequence dependent allelic variation in chromatin accessibility exists including imprinted genes, the inactive X-chromosome, and random monoallelic accessibility among others, but is likely restricted to specific genes^{12–14}.

8. Lines 567-571: “we here provide evidence that a substantial fraction of fixed genetic changes between species likely arose from highly dynamic and cell- type specific cis-acting regulatory effects. Second, our results provide direct evidence of increased fixation of regulatory changes in later spermatogenesis, which has been proposed previously in other vertebrates”? What does “increased fixation of regulatory changes” mean? How do the authors know they are fixed, as they may be just as variable within *Mus musculus* and *Mus castaneus* populations as they are between the two strains.

We agree that the word fixed was used loosely in this context, and so we removed it.

Reviewer #4 (Remarks to the Author):

Reviewer report for “The dynamic genetic determinants of accelerated spermatid evolution” by Panten et. al. 2023

In this article, the authors have used single-cell RNA and bulk-ATAC sequencing technology to examine gene expression and chromatin accessibility changes occurring during spermatogenesis. They compare gene expression in F1 males to males from parental (F0) strain background. They then go onto classify regulatory mechanisms, i.e. cis- versus trans- acting effects. A key part of the work that follows is based around this classification. They describe a large number of transcriptional changes that occurs during spermatogenesis, and ascribe this to both cis- and trans-effects on gene regulation. They state that cis- effects were generally more pronounced than trans- effects overall, and this also seen in cell sub-types. They detect increased transcriptional divergence in round spermatids stages, and from modelling suggest that the basis for this is due to allelic imbalances that exist within species.

In essence I believe the study is motivated to address how differences in gene expression levels between genetically distinct individuals, is driven by these differences in cis- and in trans-. Leveraging on single-cell RNA-seq performed on spermatogenic cells (which have a unidirectional differentiation dynamic), the authors reconstruct a trajectory of differentiation and further make comparisons between F1 males and genetically distinct parental mouse lines to ascribe regulatory effects.

Over the course of revision, the authors have including new supporting ATAC-seq data / analysis, and have revised the manuscript which supports their findings. The study is technically solid, and experiments and bioinformatic analysis have been performed to a high standard. The datasets generated will also be very useful to a number of research areas (e.g. genomics and genetics more generally). Overall the article is worthy to be considered for publication in Nat Comms.

We thank the reviewer for this assessment. Indeed, our main interest is to quantify to which extent genetic effects in *cis* and *trans* caused by the same polymorphisms vary across different cell types (and during differentiation), a major current theme in (human) genetics¹⁵. Using F1 mice as a model, we can assay these effects comprehensively genome-wide, and extend this analysis to *trans*-effects, which are difficult to detect in classic eQTL studies. Our main results are that *cis*-effects vary pervasively across cell types, and that *trans*-effects are comparatively rare. Finally, we observe that dynamic *cis*-effects are strongest in spermatids, whose transcriptomes are known to evolve particularly quickly.

Main suggestions:

1. I note that the manuscript has already been reviewed, that significant feedback from referees was already provided, and I would be broad agreement with the comments. My view is that the manuscript is generally written well, however it is pitched at a very high level. The scope of the work is vast and the subject area is complex. Nevertheless, I did find the text and some figures difficult to follow and there was also the lack of a simple message in the sub-sections. I found myself re-reading sections in order to understand.

This point was raised similarly by the editor, as well as Reviewers 1 and 2. We have heavily revised sections of the current submission to more clearly define the terminology and more concisely present the results. These revisions can be found as blue text, making up a substantial portion of the manuscript.

I believe that it may also be helpful to reorder the supplementary figures to how they appear in the text to make them easier to refer to for the general readership.

We have implemented this helpful suggestion.

2. Inclusion of a html notebook and / or relevant code of how the analysis were performed be helpful for better understanding reproducibility.

We agree that computational reproducibility is important and we have added two notebooks showcasing critical pieces of analysis regarding the detection of dynamic *cis*- and *trans*-effects to the github ("Demonstrations"). Also, all R notebooks containing the code to reproduce all figures are available (https://github.com/PMBio/ase_spermatogenesis).

3. Reviewer #3 comment: "It's also curious though that the authors identify many *cis-trans* genetic elements in spermatids although these cells are transcriptionally quiescent. Do these mutations affect somehow RNA stability since these populations are transcriptionally quiescent?"

- it is important that the authors explain this better in the text, as a major thrust of the paper centres around accelerated spermatid evolution, perhaps a sub-section under "Dynamic changes in *cis*-acting genetic effects on transcription across sperm differentiation". NB - The author revision note refers to a revised Fig.2d which I could not locate.

We thank the reviewer for highlighting this point - as it stands our data cannot directly distinguish between allele-specific expression caused by cell type-specific action of genetic variants on transcriptional regulation versus RNA stability. We have performed analyses which suggest that both regulatory effects and RNA stability may contribute, and revised the corresponding Results section (Lines 270-278, 288-

297). We also added another sentence to the closing paragraph to highlight the topic further (see Discussion, Lines 561-563).

Figure 2d refers to the heatmap in the main Fig 2, which shows that many genes change their allelic balance while differentiating from round to elongating spermatids.

4. I could not work-out how Fig.1g was generated, nor what it is trying to convey.
- relatedly, from the RNA-seq point of view, it would be helpful to expand or highlight a few more genes in Supp. Fig 1f for the benefit reader.

To improve the story flow, we have now moved this figure panel to the supplement. In Fig1f, we have highlighted more marker genes.

5. Lines 269 – 279: I could not follow this paragraph – the sentence “The two largest clusters (cluster 1 and 2; collectively covering 45.61% of all genes) showed mirror-image changes to allelic imbalance....” appears a little misleading. The percentage stated (45.61%), at least how it appears on the heatmap, appears to be high, and an x-axis label is not shown. Does the x-axis refer to individual cells ordered in pseudotime?

We thank the reviewer for highlighting this point. The percentage refers to the fraction of genes in the different clusters, each of which has a specific pattern of change in allelic imbalance (see y-axis). The x-axis represents a pseudotime and the heatmap shows interpolations of allelic imbalance for multiple genes (see **Fig2b**). The x-axis therefore does not represent individual cells, but 100 sampled points in pseudotime. We have added a corresponding label to the x- and y-axes to make this clearer, and also explain it in the figure legend. We have further added the proportion of genes between each cluster; the whitespace we have inserted between clusters may visually distort the fractions.

5. Lines 465 - 467: Please can you point the reader to the figure that supports this sentence.

We are now citing Figure 5 in this paragraph.

Minor:

1. Line 491: suggest removing the word “Briefly”, as it makes it sound like the methods section.

We have replaced all instances of this word in the results section.

1. Murat, F. *et al.* The molecular evolution of spermatogenesis across mammals. *Nature* (2022) doi:10.1038/s41586-022-05547-7.
2. Soumillon, M. *et al.* Cellular source and mechanisms of high transcriptome complexity in the mammalian testis. *Cell Rep.* **3**, 2179–2190 (2013).
3. Shami, A. N. *et al.* Single-Cell RNA Sequencing of Human, Macaque, and Mouse Testes Uncovers Conserved and Divergent Features of Mammalian Spermatogenesis. *Dev. Cell* **54**, 529–547.e12 (2020).
4. Kopania, E. E. K., Larson, E. L., Callahan, C., Keeble, S. & Good, J. M. Molecular Evolution across Mouse Spermatogenesis. *Mol. Biol. Evol.* **39**, (2022).
5. Wittkopp, P. J., Haerum, B. K. & Clark, A. G. Evolutionary changes in cis and trans gene regulation. *Nature* **430**, 85–88 (2004).
6. Goncalves, A. *et al.* Extensive compensatory cis-trans regulation in the evolution of mouse gene expression. *Genome Res.* **22**, 2376–2384 (2012).
7. Tirosh, I., Reikhav, S., Levy, A. A. & Barkai, N. A yeast hybrid provides insight into the evolution of gene expression regulation. *Science* **324**, 659–662 (2009).
8. Findley, A. S. *et al.* Functional dynamic genetic effects on gene regulation are specific to particular cell types and environmental conditions. *Elife* **10**, (2021).
9. Cuomo, A. S. E. *et al.* Single-cell RNA-sequencing of differentiating iPS cells reveals dynamic genetic effects on gene expression. *Nat. Commun.* **11**, 810 (2020).
10. Elorbany, R. *et al.* Single-cell sequencing reveals lineage-specific dynamic genetic regulation of gene expression during human cardiomyocyte differentiation. *PLoS Genet.* **18**, e1009666 (2022).
11. Floc'hlay, S. *et al.* Cis-acting variation is common across regulatory layers but is often buffered during embryonic development. *Genome Res.* **31**, 211–224

(2020).

12. Cleary, S. & Seoighe, C. Perspectives on Allele-Specific Expression. *Annu Rev Biomed Data Sci* **4**, 101–122 (2021).
13. Loda, A., Collombet, S. & Heard, E. Gene regulation in time and space during X-chromosome inactivation. *Nat. Rev. Mol. Cell Biol.* **23**, 231–249 (2022).
14. Reinius, B. & Sandberg, R. Random monoallelic expression of autosomal genes: stochastic transcription and allele-level regulation. *Nat. Rev. Genet.* **16**, 653–664 (2015).
15. Kim-Hellmuth, S. *et al.* Cell type-specific genetic regulation of gene expression across human tissues. *Science* **369**, (2020).

REVIEWERS' COMMENTS

Reviewer #4 (Remarks to the Author):

Thank you for the request to review this article once again.

The manuscript has been revised significantly and is much clearer, and the authors have addressed the concerns raised. They have also included notebooks / scripts, that which will aid re-analysis / reproducibility.

There are still one or two instances where sentences appear confusing and these should be revised – e.g. Line 206-207 “This suggests continuous changes of allelic regulation across cell states, beyond discrete cell types”. What exactly do they mean here? If including, please define cell-state changes and cell-type changes.

They have mentioned some limitations to their single-cell based approach in the Discussion. I would hence advocate a short ‘Limitations of this study’ section to specifically mention / discuss these.

Please find our responses to the Reviewer's comments in this file. We have attempted to structure our response by directly commenting on individual points raised.

Key:

- Reviewers' comments
- Our response

REVIEWER COMMENTS

Reviewer #4 (Remarks to the Author):

Thank you for the request to review this article once again.

The manuscript has been revised significantly and is much clearer, and the authors have addressed the concerns raised. They have also included notebooks / scripts, that which will aid re-analysis / reproducibility.

We thank the reviewer for this assessment.

There are still one or two instances where sentences appear confusing and these should be revised – e.g. Line 206-207 “This suggests continuous changes of allelic regulation across cell states, beyond discrete cell types”. What exactly do they mean here? If including, please define cell-state changes and cell-type changes.

We have removed this unclear statement.

They have mentioned some limitations to their single-cell based approach in the Discussion. I would hence advocate a short ‘Limitations of this study’ section to specifically mention / discuss these.

We agree that it is important to openly discuss the limitations of our study. We have consulted with the Journal and since Nature Communications does not usually feature such a section, we will leave this section in the Discussion, which states the main limitation:

“However, the detection sensitivity of current technologies is lower than bulk RNA-seq and is focused on the 3’ end of genes, thus limiting our analyses to more robustly expressed genes containing allelically resolvable genetic variants.”

On the other hand, we increase the sensitivity of our analysis through increased resolution at the single-cell level.